# Commodification of Healthcare—Patient Perspective: A Cultural-Class Inquiry of Patients’ Experience in Public–Private Systems in Israel

**DOI:** 10.3390/ijerph22101489

**Published:** 2025-09-26

**Authors:** Ram Yehoshua Adut, Nadav Davidovitch, Dani Filc

**Affiliations:** 1School of Behavioral Sciences, Netanya Academic College, Netanya 4210002, Israel; 2School of Public Health, Faculty of Health Sciences, Ben Gurion University of the Negev, Beersheva 84105, Israel; nadavd@bgu.ac.il; 3Department of Politics and Government, Ben Gurion University of the Negev, Beersheva 84105, Israel

**Keywords:** commodification, class, ethnicity, healthcare, self, neoliberalism

## Abstract

This study discusses subjective aspects of the commodification of healthcare from an ethno-class perspective using narrative analysis of patient stories. We hypothesize that the objective social hierarchy of resources, together with a certain degree of individual agency, structure the patients’ strategies of coping with the public–private “maze” of the healthcare system. The findings show different coping strategies indicating three different ethno-class ‘patient-selves’: The dominant ‘Neo-Liberal Self’, prevalent among the upper middle-class (mostly Ashkenazi Jews) that expresses contempt of the public system, and an individual hero-quest story maneuvering between the private and the public. The ‘troubled’ patient-self of the low-middle and working classes (mainly Mizrahi Jews and Arabs) also expresses negative impressions of the public system, but it is drawn to sadness, fear of being lost, and a longing for a lost ‘logic of care’. Finally, a ‘communal alternative self’ among the Arab lower classes seeks personal solutions through social networks that include local health providers while crossing barriers between private and public sectors. All selves show some degree of neoliberal values, but the first ‘patient-self’ implies a sense of social mastery while the other two attest to the agency and even resistance of patients facing structural barriers and scarce resources.

## 1. Introduction

Israel’s national health insurance and universal service delivery system has long been a source of pride for many—dating back to its foundations prior to 1948, through the immense challenges of mass immigration in the 1950s and 1960s, and culminating in the enactment of the National Health Insurance Law in 1994. It appears to be rooted in a collective imagination shared by many as a pillar of Israeli social welfare and a reliable safety net. Most services are delivered by non-profit health maintenance organizations (HMOs), and public hospitals remain largely accessible and open to all Israeli residents [1].

However, since the late 1990s, this predominantly public universal system has faced growing challenges and has undergone a series of policy reforms that introduced elements of privatization and commercialization at its core. For example, although all Israeli residents continue to be covered by the national insurance scheme, non-profit HMOs now offer supplementary voluntary insurance that provides additional services and choices for a fee. In parallel, for-profit commercial insurance companies offer significantly more expensive private insurance options [2].

These and other changes (outlined below) have resulted in a complex hybrid of public (non-profit), private, and semi-private services and insurance arrangements. Academic and professional communities have long debated these developments—torn between the values of “consumer choice” and “freedom” on the one hand, and “care” and “equality” on the other. Over the past three decades, many of the most significant reforms have been criticized as “neo-liberal” in nature, associated with deregulation, privatization, and the introduction of out-of-pocket payments—even within the core of the public system [2].

Scholarly research on these issues has primarily focused on policy analysis and the macroeconomic dimensions of healthcare. The subjective preferences of “the public” often remain a constructed image shaped by various stakeholders. Approaching this subject from a research perspective is far from straightforward. The ways in which individuals—positioned differently within society—navigate the “maze” of public and private services remain relatively underexplored or vague. Existing studies have largely relied on quantitative surveys and polls to assess public attitudes and opinions [3].

While those are valuable studies, they are not able to capture the depth of meanings, motivations and lived experiences that characterize Israeli society with its complex hierarchical mosaic of social groups.

The present paper—and the broader study from which it is derived—reflects this complexity. The broad research project, of which this paper is one part, examined the intersection between private and public healthcare along four dimensions: the first involved semi-structured interviews with policymakers [4]; the second, interviews and surveys with physicians [5]; and the third, a large-scale statistical analysis of a telephone survey of patients [4]. The first two studies yielded relatively clear findings. The third, while offering important insight into levels of public trust and choice, yielded more ambiguous results regarding how different social groups experience the public/private healthcare interface. Later, the same author (Niv-Yagoda) presented another cross-sectional telephone survey among a representative sample of Israeli adults [6].

The quantitative surveys’ findings may suggest that the analytical tools employed need reconsideration. We posit that coping strategies and the meanings attached to healthcare experiences may be socially shaped by group belonging and hierarchical positions, but that these dynamics must be investigated using more nuanced, qualitative methodologies, at least in the first analytical exploration. Qualitative research, we argue, should be the necessary foundation upon which broader, quantitatively-driven research programs can be built.

Our second ground-line in this paper regards the neo-liberal social climate. We hold that the commodification of everyday life is a central characteristic of neoliberalism and it is highly effective in present-day Israel as it is in other developed and developing countries [7,8,9,10,11]. As a recent study showed, the European Union developed a New Economic Governance framework, that actively promoted commodification of health care in countries with different health care systems: conservative regimes such as Germany, Mediterranean such as Italy or liberal such as Ireland [12,13]. Presumably, the consequences of commodifying welfare services, such as inequality and inefficiency, require groups and individuals to develop coping mechanisms to properly enjoy and have access to the commodified service [14,15]. Following these processes, this paper uses Israel as a case study to discuss how different ethno-class groups determine the ways that individuals navigate a semi-commodified healthcare system in the neoliberal age.

These are two foundational premises of the study; however, they do not lead us to formulate a priori hypotheses about specific social groups or the precise ways in which the neo-liberal cultural climate is assumed to influence their attitudes and strategies towards public–private health dilemmas.

This open-ended attitude led to the adoption of the following framework: a life narrative analytical approach combined with a grounded theory–inspired methodology (see the methodology section for a more detailed discussion). The life narrative approach aims to uncover and highlight patterns of interpretation, coping strategies, and the broader frameworks through which individuals make sense of their life circumstances and decisions. The literature on narrative methodology encompasses a wide range of emphases, including the reconstruction of meaning structures, the analysis of social relationships, and the amplification of subjective perspectives [16,17].

The grounded theory approach is primarily methodological, guiding both data collection and analysis. It requires the suspension of hypotheses until the later stages of analysis, allowing patterns and categories to emerge from the data itself [18,19].

In the following sub-sections, we begin by defining the concept of ethno-class and then provide a brief overview of Israel’s ethno-class structure. The second section describes the Israeli healthcare system(s) and the complex interrelationship between its public and private sectors in the context of neoliberal reforms. The third section outlines the scholarly literature on the subjective-cultural aspects of health commodification in developed countries. The core components of the paper present the methodology used, the findings and their possible explanations.

### 1.1. Ethno-Classes in Israel

We define class as a social group, or a web of social groups sharing common material conditions (the relationship to capital) that are linked into power relationships with other classes. Class is also defined by shared elements: shared space, practices of consumption, partially shared culture, forms of individuation and, sometimes, common political worldviews. All coalesce in the way a certain social group will define the boundaries between itself and others [20].

In Israel, social stratification is characterized by ethno-classes. We understand the concept as the interaction between ethnicity and class, in such a way that ethnic identities strongly determine the individuals’ position in the production process (i.e., class belonging), and class positions become an element in how the ethnicity defines itself [21].

Addressing Israeli society from the 1950s and through the 1990s, researchers such as Rosenfeld and Carmi [22], Bernstein and Swirksi [23] and others have shown that Israel’s economic development combined with its citizenship regime resulted in a three-tiered ethno-class structure. Most Israeli Arabs (the Palestinian-Arab minority since 1948) occupy the lowest echelon, Mizrahi Jews (originated from Asian or North African descent) are associated mostly with the lower-middle status, and Ashkenazi Jews (originated from European or American origin) are at the top of the ethno-class structure. “Mixed” Ashkenazi and Mizrahi families were found to be quite upwardly mobile, at least in academic education [24]. Russian immigrants from the former USSR are still located in the lower layers of Jewish society but some have been mobilizing upwards, and finally Ethiopian Jews are at the lowest position among Israel’s Jewish community. The constitution of ethno-classes in Israel included differences in the position in the production process, income, education and spatial segregation.

The neoliberal changes that Israeli society underwent during the 1990s and 2000s, such as the weakening of the ethno-republican ethos and the expansion of higher education, modified this three-tiered ethno-class structure. As shown by Cohen and Leon [25] in their seminal research, a new Mizrahi middle class emerged, distinct from the Ashkenazi middle classes but also from the Mizrahi working class in terms of cultural, social, and economic capital. Additionally, a new middle class emerged among Arab Israelis [26], making the Israeli ethno-class structure more complex: A middle and upper-middle class comprised mostly of Ashkenazi Jews, a Mizrahi new middle class, an Arab Israeli new middle class, and a lower class characterized by diverse ethnic subgroups (Mizrahi, ultra-Orthodox Jews, Russian and Ethiopian Jews, and Arab Israelis). Moreover, the lowest echelon is composed of daily workers who are Palestinians from the Occupied Palestinian Territories and by migrant workers, not covered by the national health insurance law. As a result, they have a completely different avenue of navigating the healthcare system which lies outside the scope of this research.

### 1.2. The Israeli Healthcare System

The Israeli healthcare system guarantees access to a universal basket of services which every resident is entitled to, as established by the National Health Insurance Law (NHIL). Two mandatory taxes—health and income—are the main funding sources. However, in reality, Israel’s healthcare ‘system’ operates as a mixture of public and private insurance(s) as well as certain privatized services, either privately owned or financed [5].

The Ministry of Health (MOH) is responsible for the planning and supervision of the healthcare system, which includes hospitals and four public, non-profit health maintenance organizations (HMOs). The HMOs provide the services within the ‘healthcare basket’ as defined by law and adjusted by regulations, including primary and secondary care, and finance. While many hospital beds are operated by the state, HMOs sometimes provide hospitalization services. Alongside and intermingled with the public system is a growing private sector.

Over the last three decades Israel has experienced a steady, gradual process of privatization of financing and ownership of healthcare accompanied by a cumulative erosion of publicly provided healthcare services. Rising costs, driven by demographic changes and rising healthcare prices were not matched by increased public financing, causing a cumulative deficit of about 26%, a shortfall of about NIS 20 billion (5.4 billion USD) in the HMOs’ budgets [27].

Neoliberal reforms in the Israeli healthcare system included a significant reduction in public funding—from covering approximately 75% of total health expenditures in the early 1980s to around 60% by the 2000s. Additional measures included the abolition of an earmarked employers’ tax, the introduction of copayments for medical imaging and specialist consultations, the outsourcing of services to private providers, and the corporatization (trustification) of state-owned hospitals. A crucial aspect of commercialization is the expansion of private supplementary health insurance sold by the public HMOs (see below).

To compensate for the diminishing public budget, the government introduced significant increases in copayments for medications and specialist care. Moreover, since the 2000s, budget constraints have pushed hospitals to develop a range of private initiatives to increase revenues. Healthcare expenditures of total household spending rose from 3.8% in 1997 to nearly 6% already in 2014, which has impeded lower income people’s access to healthcare [28]. This rate remained at 6% in 2022 [29]. The increase in households’ health expenditure is attributed primarily to out-of-pocket payments (copayments, visits to private doctors) and to purchasing and using supplementary or commercial insurance.

A three-tiered system has evolved between types of insurance coverage: public-mandatory, public-mandatory and supplementary (sold by HMOs), and finally, public-mandatory and private commercial. Here is a detailed description of the three layers.

In Israel, the term “HMO” (Health Maintenance Organization) refers to one of four non-profit, government-regulated health funds known as *Kupot Holim*: Clalit, Maccabi, Meuhedet, and Leumit. Unlike HMOs in the United States, these are not private, for-profit entities, but rather public institutions that provide universal health coverage under the framework of the National Health Insurance Law. All Israeli citizens and permanent residents are required to enroll in one of these HMOs, which are financed through a combination of a national health tax and government subsidies. This structure guarantees access to a standardized basket of health services for all residents, regardless of income or health status.

However, the actual situation is highly complex. Supplementary insurance serves as a particularly illustrative case. This is a semi-private insurance, regulated by law and sold voluntarily by the public HMOs to their members who already have universal basic insurance. It covers services not included in the basic health basket, such as dental care, alternative medicine, private hospital rooms, and faster access to specialists. The supplementary insurance is open to all HMO members, regardless of pre-existing conditions, but is based on age and not income, turning it into a regressive source of funding. Importantly, this insurance is provided by the same non-profit HMOs that administer the public healthcare system, not by commercial, for-profit insurers. This is actually turning it into one of the examples out of many other private-public mix that are quite common in the Israeli healthcare system, leading to many market failures and inequities. However, the main driver to purchase supplementary insurance is for services which are already included in the mandatory public insurance, including the ability to choose surgeons and, primarily, to shorten waiting times for services guaranteed by public insurance [30].

Commercial health insurance in Israel is provided by private, for-profit insurance companies. These policies offer additional coverage beyond both the basic public health basket and the supplementary insurance provided by the health maintenance organizations (HMOs). Commercial insurance may include benefits such as elective surgeries with a chosen surgeon, access to a broader range of medications, and coverage for advanced medical technologies or treatments abroad. Unlike public and supplementary insurance, commercial insurance is risk-rated—meaning that premiums and eligibility may vary according to factors such as age, health status, and medical history—and coverage is not guaranteed to all applicants (Table 1).

While the entire Israeli population is mandatorily covered by public insurance, 84% of the population owns supplementary (semi-private insurance provided by HMOs) and 57% owns the more expensive commercial, with 52% owning both (rates are still rising since 2019). Figures from 2022 show that within the lowest quintile, 33% have only public insurance. Among the Arab population, 46% have only public insurance. Most of the other members of this quintile have supplemental insurance, but not commercial insurance [32]. Thus, access to private and commercialized health insurance products is undoubtedly stratified along ethno-class lines. Availability alone does not tell the whole story. The patterns of usage of supplementary insurance vary according to ethnicity and regional factors, even among groups in the lower quintiles [33]. Patients with public health coverage are deprioritized in terms of access, waiting times, and the attending specialist’s seniority, compared to commercial or supplementary insured patients. Furthermore, most of the new services offered within public insurance are provided in the country’s central area (around Tel Aviv and Jerusalem), increasing inequalities in access between the center and the periphery [2].

The commodification of healthcare has two main manifestations: institutional, seen by the expansion of private services, and subjective, by the adoption of a consumerist approach among healthcare service users. This consumerist attitude pervades when patients are accessing both the public and private services. Both the public healthcare institutions and the system’s users see the latter as customers who use market strategies to navigate the system, strategies which are presumably conditioned by ethno-class belonging [6,34].

### 1.3. Cultural-Class Research on Semi-Commodified Healthcare

Since the 1990s, several studies have been published in Great Britain on the patterns of use of health services, using qualitative research methods to study the increasing commodification [35,36]. Scholars in rich welfare countries soon noticed and delineated the expansion of ‘the logic of choice,’ which began to overcome ‘the logic of care,’ the founding principle of the public healthcare systems of the mid-20th century [37,38].

Australian scholar Fran Collyer and her colleagues conducted an extensive review of the sociological literature on health systems in Western welfare states with universal health coverage [39]. Across these countries, a range of reforms have been introduced under the banner of “choice,” resulting in a mixture of public and private healthcare services. These reforms have created what the authors describe as a potentially confusing “maze” for patients. However, Collyer and her colleagues found a dominant reliance on rational choice theory. Within this paradigm, individuals are typically portrayed as independent, stable, and consistent consumers who always prefer more choice over less. By contrast, the authors noted a significant lack of research focusing on patients’ actual lived experiences—their emotions, social contexts, and subjective meanings as they navigate complex healthcare systems.

The authors’ call for a shift in the research agenda—an approach they themselves have begun to pursue—draws inspiration primarily from the theoretical framework of Pierre Bourdieu, and his understanding of how social hierarchy is linked to culture.

According to Bourdieu, people’s deepest dispositions, preferences, values, and perceptions are shaped by their social group—a phenomenon he termed habitus following Norbert Elias. Habitus is a set of embodied tendencies formed through one’s social and material conditions. It reflects not only economic capital but also social and cultural capital. Bourdieu argued that the unequal distribution of these resources among groups forms the basis of a multi-dimensional stratified society that still tends to reproduce class lines s [40].

Collyer and her colleagues argue that ‘cultural capital’ (among other linguistic competence, intellectual confidence, and procedural “know-how”) is perhaps the most potent of Bourdieu’s concepts for analyzing the experience of patients navigating hierarchical healthcare systems. Cultural capital is unevenly distributed along socio-economic lines—not only between the wealthy and the poor but also between the educated and uneducated, and between central and peripheral social groups.

The third pillar of Bourdieu’s framework highlighted by Collyer is the concept of the ‘field’—a structured social space in which actors and institutions occupy different positions based on their accumulation of various forms of capital. Each field operates according to its own logic and hierarchy, redistributing power and capital unequally. In this sense, the healthcare field can be viewed as a contested space where corporations, state institutions, and patients from diverse social backgrounds compete for influence and resources.

Broadly following this line, based on ethnographic observations in the Netherlands, Annemarie Mol provided a phenomenological analysis of ‘choice’ as the main symbolic axis of a consumerist discourse which commodifies the patient. Mol postulated that while choice carries an extra burden on patients who are upper-middle class, namely, those who are highly educated with high ‘Cultural Health Capital’ (CHC) [41], it imposes a greater burden on patients from the lower echelons of society, with lower CHC. The latter is more likely to get lost in the health ‘maze’ created by increasing waiting times, access barriers and the complexity of the public–private mixture [39].

Jannette Shim’s analysis is perhaps the most extensive theoretical elaboration of Bourdieu’s terminology. Shim claims that the entire health system in the United States is a network of institutionalized arrangements, interlinked with the uneven allocation of cultural health capital (CHC) among patients. CHC is a repertoire of: “cultural skills, verbal and nonverbal competencies, and interactional styles that can influence health care interactions at a given historical moment.” [41] (p. 2).

However, Shim’s interpretation of Bourdieu, despite its emphasis on structural conditions, falls back on the individual as the basic unit. It seems to underscore the inner-group dynamics and concentrate on the crucial moment which an individual faces. From our perspective, it does not offer a flexible-enough theory to analyze the complex ways—both material and symbolic—with which specific social groups make up their members’ experiences in accordance with the group’s objective position in the socio-economic hierarchy. Other American scholars, like Jeniffer Malat [42], took CHC to study the uneven allocation of health on a group basis, stressing the racial aspect. But this aspect of ethnic distinction is quite different in Israel.

Even more in line with the Australian research agenda outlines above, Harley et al. [43] suggest that the seemingly individual isolated action (choice) is in fact rooted in social class structure, where different types of (Bordieusian) capital are (re)created and interplayed. They state that the healthcare field “contests between the dominant ‘position-takings’ … those of the corporations of capitalism … the capitalist state … and those of subordinate actors.” [44] (p. 690). Applying this approach, Willis and Lewis found that the choice discourse penetrated deeply into diverse social strata and even among the disadvantaged echelons [45].

The present study is informed and inspired by the above scholars and research agendas, but some fine theoretical adjustments must be made to account for the Israeli situation as we observe it. Specifically, we add two requirements that we believe contribute to understanding the specific ways in which class and ethnicity interact with the commodification of health care. Firstly, it seems to us, as both native members and scholars of Israeli society, that the concept of “choice,” while central in academic and professional debates, is not necessarily the dominant cultural code across Israeli social arenas. Therefore, when we approach the interviews, we broaden the analytical lens: our question is more general than the issue of choice alone. We aim to explore the ways in which experiences shape subjectivities as users or “consumers” of health care, particularly in contexts beyond the framework of “choice”—such as that of Arab citizens seeking services within the public health system under conditions where choice is limited or absent.

Secondly, we place greater emphasis on the assumption that the complex interactions within each social group shape not only the individual patient’s choices and health care conduct (CHC) but also the broader ethno-class repertoire of meanings and actions that emerge within complex health systems.

Lastly, we are highly aware of the potential contribution of Bourdieu’s “Field” theory of structural power relations [46]. However, this paper focuses on collective-individual experiences and strategies from the perspective of the agents, and “Field” analysis would exceed its scope. External constraints led us to concentrate our full ethnographic attention on the patients’ narratives in a dynamically commodified health system to elucidate issues of subjectification and agency and to leave the question of health system structure as Field for further research.

The above discussion demonstrated the need to redefine our basic unit of analysis. This we accomplish by using the term ‘a classed self’, following the theoretical and empirical work of Beverly Skeggs [47]. Like Skeggs, we assume that the classed self is the product of social discourse where economic, social and cultural capitals are at play and the whole structure is determined by class logic. Skeggs’s scheme is inspired by both Neo-Marxism and by Bourdieu, but doesn’t fully comply with either. In her theoretical work, Skeggs demonstrates that even the great critical thinkers like Bourdieu are in fact rooted in a hegemonic concept of a classed self, namely, a white, European, upper-class male [47].

Theoretically, Skeggs expands the concept of the ‘habitus by arguing that in social reality the classed self reflects the main power lines in society (class, gender, ethnicity) but it is mainly understood as a product of cultural interactions, that is, of power relations which exist through symbolic representations and generally through discourse. In her empirical work, Skeggs showed that symbolic force is indeed drastically repressive in class terms, but it can also be challenged from below. In her work on the repressive representation of working-class women, Skeggs postulated that in British class society two or more “selves” can evolve through complex power relations immersed in hegemonic mass culture. On the one hand, this analysis is a demonstration of class domination; on the other hand, it delineates a sort of subversive agency by subordinated groups and specifically by women [48]. Although this concept is arguably inspired by the Bordieusian habitus and by the notion of symbolic violence, it implies a much stronger sense of dialectical agency on the part of the agent (individual and collective), which can produce diverse coping strategies and even mobility and empowerment.

In summary, the classed self’s basic contours are determined by the group’s objective material and symbolic conditions in the class-ethnic hierarchy. While it is a product and vehicle of repressive conditions, it can also become the agent of certain coping and subversive practices. Finally, one should note that even these subversive and coping individual practices are themselves basically a collective construction which the individual adopts and adjusts. The relation between the collective repertoire and the individual action resembles a standard melody that a jazz musician improvises on.

## 2. Materials and Methods

This study aims to assess the influence of the neoliberal discourse of commodified health among different ethno-classes by exploring patients’ perspectives. The analysis is based on narrative interpretation of semi-structured interviews with 20 respondents, supplemented by several informal conversations with key informants. The respondents were approached using several ‘snowballs’ within different regions in Israel.

The sampling approach was tailored to the study setting. The three authors drew on their long-standing relationships—jointly and individually—with several grassroots and local advocacy NGOs working in health promotion across different regions and ethno-class communities. These NGOs varied in orientation: some were public-health and community-based organizations, others more advocacy-focused—but all were non-profit and had no commercial ties to public or private health providers (apart from occasional voluntary community initiatives). Local leaders of these NGOs suggested potential interviewees from their communities based on direct acquaintance. The leaders were told only that the study examined the public–private health interface, a contested topic even within health advocacy. Eligibility criteria for all respondents were adulthood and current or past health issues. Thereafter, contact with the chosen respondents was conducted directly between the research team and each prospective participant. To avoid over-reliance on any single network, each referral route yielded no more than four respondents. Some leaders were re-approached in the second phase for further discussion on the themes (see below). Each semi-structured interview began by asking the respondent’s occupation, education, age, and their general ethnic attributes. It then moved to questions regarding personal experience, values, and views on the mixed private-public healthcare system.

Our approach is a combination of life-narrative analysis with grounded theory inspired methodology. This means that, although we acknowledged certain assumptions regarding neoliberal conditions and the ethno-class structure of Israeli society, we did not impose any prior assumptions or pre-formulated hypotheses about the specific coping strategies or meanings our interviewees—or their respective ethno-class groups—might attach to their health-related experiences at the intersection of public and private care. While we were interested in uncovering ethno-classed selves, their specific cultural content—and even the question of whether such patterns could be persuasively generalized —was deliberately left open to emerge inductively from the data [19]. All the interviews were performed between 2020 and 2021, more than two years before the October 7th attack and the beginning of the war in Gaza.

The texts were analyzed using the ATLAS program to identify common themes and values, narrative patterns, and interpretation of reality. In line with the tradition of narrative analysis, we assumed that an individual account of a seemingly isolated personal event might contain the seed of a broader ‘key plot’—a collective story that encapsulates the experiences and interpretive repertoire of a wider group. Such narratives, we believed, could be elaborated into the concept of an “ethno-classed self” [16,49].

The initial coding of the data was followed by the development of first-level hypotheses or generalized codes. This step was conducted independently by two team members, after which the findings were discussed collectively by all three researchers. Only those generalized themes that received full consensus were advanced for further elaboration. This process continued until we felt confident in identifying and articulating the existence and content of distinct “ethno-classed selves.” Following this stage of analysis, we returned to several key informants (community or local NGO leaders) to present and refine these emerging generalizations. This phase was inspired by the principle of “theoretical sampling” as articulated by grounded theorist Kathy Charmaz [19].

The study was approved by the ethics committee at Ben Gurion University Faculty of Health Sciences (19/2020). All participants received a guarantee of full confidentiality from the interviewer and gave recorded consent of participation prior to the interview. The interviews were all conducted in Hebrew by two qualified researchers (PhD), one of the authors and another who is not one of the authors of this paper.

See Table A1: respondents by name, gender, age, profession, education, residence, ethnicity, class position.

## 3. Results

The interviews showed that different ethno-classes presented distinct strategies to navigate the health care system.

### 3.1. Upper Middle-Class—A Hero-Quest

Ronny is a 77-year-old Jewish-Ashkenazi insurance agent with an academic degree, residing in Israel’s urban center, which is an upper-middle class space:

“I had a back problem. I ran around in circles [and went] to the best consultants. Nothing relieved me from the suffering… until I ran into a young doctor who just came with a certain specialty in surgery after studying in Pittsburgh [USA]. He examined me. I handed him an MRI I took, and he told me, ‘Dear Sir, I am not working with the public system on this matter because they don’t want me there. You are invited to the private system.’ In two minutes and twenty seconds, all the documents from the insurance company were transferred to the hospital. I didn’t pay a single penny for the surgery. He operated on me, and I danced my way out of surgery.”

Ronny said that he trusts the public doctors, but he despises the system, citing bureaucratic barriers, lack of credit for talent (not wanting the ‘young doctor’), low availability, and low quality of treatment. His story is an individualistic one of salvation, a sort of hero quest in which he is the hero seeking a solution. He zigzagged between public and private until eventually finding his solution in the private sector.

Other respondents, who can be characterized in Israeli ethno-class terms as Ashkenazi upper-middle class subjects, tell similar stories with individual variations. While not all are success stories, they always express harsh critique and even contempt for the public system, and a very strong assertion of individualism and the existential need to turn to private providers. Ultimately, their story is a story of an individual patient who succeeded in navigating a very complex system using personal resources.

Ada’s occupation is lower middle-class, but she is a kibbutz member, thus culturally belonging to a mostly Ashkenazi upper-middle-class community in ethno-class terms. She feels obligated to praise the public health system, but her stories ultimately deliver a totally different message. She shares with Ronny a deep distrust of the public healthcare system and in times of need she turned to private insurance which the kibbutz (despite its past socialist ideology) provides. Ada tells the following story about her mother:

“My mother has a problem with her eyes … She was blind in one eye and then, in the other eye, she suddenly had some kind of hemorrhage, and she couldn’t see. We went to doctors in X [a public hospital]. They tried and said, ‘There’s nothing we can do…She won’t see anymore.’ She sat at home for three months … no TV, no books, nothing. Four walls and that’s it. … We were looking for a [private] specialist doctor [and] we found [someone] through the internet. When I suggested [to the public doctors] [give her] Avastin shots, do something! They said, ‘No, she doesn’t meet the standard, she isn’t entitled to it’. We went to her private doctor. [After] three injections, [my mother could] see and read. Since then, they [the public hospital staff] are erased to me.”

Ada despises the public hospital’s ophthalmology department and is unable or unwilling to accept the allocation criteria which the public healthcare system uses.

### 3.2. Low-Middle and Working Classes—Barriers and Humiliation

Ira, a 72-year-old retired bookkeeper who immigrated from the Former Soviet Union (FSU) in the 1990s, lives in a peripheral city. She is chronically ill with many medical needs:

“I think that the health system does not work for the… ordinary person. Our health system only works [for] people who have money. If you have money, you will get really good quality treatment. If you don’t have money, well, that’s where the story starts. Queues, for example … you can’t get [appointments] easily. You can get it with ‘special favors’ [‘protektzia’] or privately. Money. Pay money [for] surgery. That’s what we did.”

Like Ronny and Ada, Ira appealed to private medicine for a solution. Unlike Ronny or Ada, however, her story is not heroic, and her attitude is not contemptuous. She told several stories of hardships and despair, and eventually she expressed nostalgia for the days when she felt she could just trust the system.

Rivi is a Mizrahi woman, a retired factory worker from a small town in the periphery. She also began her narrative by mentioning money:

“When it’s about money, then everything is OK. But when you have to go to a professional doctor in the [HMO], then it’s…problematic because the waiting is long … This is very bad, and those without money are [lost]. They fall through the cracks. I’m telling you—I went to a gastroenterologist [using supplementary insurance, choosing a private specialist]. I hardly sat down, and he already called my name. Isn’t it surreal? It’s very surreal. And meanwhile many people wait in line [in the public system]. Surreal!”

This narrative clearly centers on the commercialization of insurance and the perceived benefits it offers—namely, a kind, personal, and prompt response. It exemplifies how supplementary insurance is utilized by a lower-middle-class individual who has spent most of her life within the confines of the public system. Rivi’s account testifies to the formula “health equals money,” yet her tale is not about private medicine’s victory which she celebrates. She doesn’t express any contempt. The whole matter is rather sad and bewildering in her eyes.

Yossi, a middle-aged Mizrahi locksmith from a relatively poor Jewish town in the periphery told of his experience in the public system after a work injury:

“I haven’t worked because of my ankle for almost six months … So, here there are two doctors: one is blabbering, and the other listening. The first one talks to the other … Really, they are not interested in treating you.…Eventually, he writes something to himself [and says to me], ‘Okay, take this [medicine] and leave.’ I didn’t like one bit’ …”

Later, Yossi needed an operation, but he refused:

“Why? Because I was afraid that they will bring in an apprentice. Sometimes the doctor stands still and brings in an apprentice. And I don’t want him to instruct [the apprentice] at my expense. I went to a private doctor because he knows. I said to him: ‘Are you doing [the surgery]?’ He told me: ‘Yes.’ And it was the most important [surgery] I’ve had … We paid what was needed to be paid and we did it.”

Yossi too speaks of a degrading experience with the (public) orthopedic surgeon:

He “Treats you but [does] not really treat you. Yalla, Yalla! Finished? Send in the next line! … It’s like a factory, a conveyor belt.”

Like others, Yossi’s story conveys no sign of victory, contempt or mockery but is moved by a frustrating experience, bordering on humiliation, and it ends by appealing to private medicine that carries a price tag.

Ja’far, a middle-aged Arab, is a construction professional, and a building team manager. His story started with a work injury:

“My worker was wounded by a nail at work … I went to the clinic. They told me: ‘You must go to [another clinic in town]; he needs trauma [care].’ … So, I took the guy and went there. At this clinic there was this disgusting doctor. He’s a trauma doctor, for cases like this, [but] he [the doctor] didn’t agree to see him [the worker] … because he stained the entire clinic with blood. [So] I took a referral letter and went to the [private] Nazareth English Hospital. And all the time, he cries that it hurts, and the blood comes out of his leg.”

Ja’far shared another story about his father:

“I remember my father, may his soul rest in peace, had lung cancer and was hospitalized in X [a public hospital] … At the end of his life, they put him in a disgusting, filthy place. …. The doctors told us there was no chance—just to wait … If he was a private [patient] this hospital would have treated him differently … He would have received all the conditions that a person deserves. [My father was] someone who had passed so many years and contributed [to society] … You know, he also worked in the National Transportation Agency, [in the] government.”

Ja’far’s story is filled with pain and anger, but it does not express contempt for the public system. While the story of his father’s hospitalization may better reflect the limitations of a public hospital due to years of underfunding, his first story exemplifies a story of disrespect grounded on the combination of classism and racism. Both stories bring Ja’far to seek private services because in his eyes, he had no other choice. Identifying with his father only intensifies the insult.

Warda, a professional academic woman, moved from the north to the south and faced the harsh social conditions in which the Bedouin live. While Warda herself belongs to the Arab mobile middle class, she serves as an informant for lower class Bedouins:

“You call to book an appointment. Your condition is difficult, and you need a diagnosis because you have recurrent infections and a swollen face, and you’re not functioning. Then you’re told that the appointment will be in six months … We will send you a letter to the post office with your appointment and it will be about six more months. [But] the postal service is lousy in the village. We don’t get mail. So, there’s no way to receive an appointment over the phone … And they say, go back to your office at your [local] clinic, but it’s lousy at the clinic. In the end, I had to take money out of my pocket and go to a private doctor [in the north]. I reached him through an acquaintance, a doctor [I] studied with at the Hebrew University … ‘You have money, [or] you don’t have money. Just go.’ I went and I paid 500 shekels. I had surgery and that’s it. The pain was gone.” Warda’s frustration concerning the public sector results from the combination between the limited capacities of the public sector especially in the geographical periphery, and structural racism as expressed in the lack of infrastructure in the Bedouin unrecognized villages.

Warda’s narrative started with complaints about the public HMO and ended in the private sector.

### 3.3. Low-Middle and Working Classes—Supplementary Insurance

The fact that a relatively significant number among the lower middle and working classes own supplementary insurance demands an examination of how they make use of this insurance and the significance they attribute to it.

Nora is an elderly Mizrahi woman from a relatively poor village (moshav) in the remote southern periphery. During her long marriage, she felt no special need for any supplementary insurance or private medicine. Only after her husband’s death, she bought the insurance, and it suddenly paid off:

“I was at an [public] ophthalmologist 18 months ago. He told me to do a cataract [operation] urgently [and] gave me a reference letter. I brought it to our [public HMO] clinic. They told me: ‘Well, we’re faxing it to X [regional HMO hospital]. They’ll send you an appointment time.’ … I’ve been already waiting for a year … [then] I received a letter in which they tell me, ‘You’re in the waiting line. When the appointment is available [we’ll call you].’

So, I rang X [a private clinic] … they made it perfect … it was patient, [people] talking nice to you.”

Further inquiry revealed that Nora’s children played a major role in her decision to buy supplementary insurance and assisted in covering the required copayment for the surgery (~2000 $). Her children are relatively upwardly mobile, third-generation Mizrahi Jews. They were very clear, Nora says, in stating the supplementary insurance is a ‘must’.

### 3.4. Arab Low-Class—Communal Networks

Jaber, a middle-aged plumber from a poor and small Arab town in the periphery, bluntly declared that ‘I like to go to private medicine.’ A closer look revealed a world of difference between what Jaber calls private and the regular meaning which mostly refers to supplementary or commercial insurance. Jaber’s route is different. It starts with the local family doctor who is also a distant relative:

“So [my] doctor tells me, ‘We have a doctor in our system [HMO] … first of all, you go to him.’ Like, he wants to save my money… So, we go to him. If it doesn’t help … then we go to a more specialist doctors.”

What Jaber casually depicts is a surprisingly complicated network which was recurrently revealed in almost all the Arab narratives. It starts with ‘our’ doctor—a close acquaintance, sometimes even a relative, who resides in the same village or town. This physician sits in the local public clinic or in a nearby hospital. Yet, they are but a station on the way forward to other doctors that the patient does not know personally. Here no one writes referral letters or official papers, the whole route seems to run smoothly and informally.

At this point, says Jaber, it might turn out that he needs another specialist, and more effort is needed to reach other doctors or services. This path might start within the public system and then cross the border to other institutes or to the private sphere. When asked about payments, Jaber reluctantly admits that some payment might be required.

Belal, a successful NGO manager, lives in his home village which ranks in the lowest socioeconomic quintile. He is undoubtedly a local leader and is able to throw light on what he terms ‘the Arab ways’—distinctive routes to get better healthcare that cross the borders of public and private. This whole network, he says, is based on personal connections among Arabs.

“I pay. Absolutely, like everyone else. Even more sometimes if [the doctor] asks. I have no problem. If you reach a critical point and you want to heal someone, you … don’t haggle over prices here.”

For Belal, this practice is simply a survival strategy deployed by an Arab minority that suffers discrimination and sometimes humiliation in the public system. The following account shows how he justifies this strategy even though he is a supporter of health equity:

“I hear about many cases of people who are dependent, very shaky. … [M]y uncle, for example. He went to [a public hospital]. After three months of waiting, he got there, and it turned out that the referral letter is missing … he said: ‘[The doctor] didn’t accept me. I have to go back [home]’… I said to him, ‘Why are you coming back? We’ve been waiting three months for this! Let’s pick up the phone!’”

Eventually Belal intervened and managed to sort things out, but his uncle was too tired and went home. The system won according to Belal’s interpretation. Belal’s story also results from the combination of the insufficient capacities of the public sector (long queues) and racism (sending his uncle back without a physician checking him).

Warda related another story about communal networks of personal acquaintances:

“My daughter suffered from stomach pains and constipation, and our [HMO] has no pediatric gastroenterologist in the [Bedouin] Negev. So, … I had to go to the French hospital in Nazareth in the north … simply because [the HMO] could not give me a close enough [appointment] date. [Here in the Bedouin Negev] there is no doctor who belongs to [my] HMO. There are no upcoming appointments. … I went to the French hospital because I knew people who were treated there. I knew someone who works there.”

Warda continued to describe some features of the communal networks among the Arab Bedouin communities in the Negev which are ranked the lowest SES in Israel. In these regions, she explained, there is a unique mixture of HMOs and the local networks. According to Warda, the HMO chooses a person who is a social figure in the local community, a sort of local leader, whose mission is to bring in people to register in the HMO and buy its supplementary insurance. He might even bring in entire families. From the patients’ perspective, this person will serve as their representative, smoothing their daily contacts with the HMO, coordinating formally and informally, weaving networks that can aid in times of medical need. This sort of person can easily cross formal barriers.

## 4. Discussion

Similar to many western countries, Israel has gradually experienced a well-documented process of healthcare commodification [2]. As in other countries, these processes have created a complex public–private ‘maze’. While most policy makers adhere to the axiom that ‘people want choice’ [50] (p. 13), our research raises questions about the deep impact of commodification.

Respondents in our research were asked generally to tell us their own personal stories about their healthcare experiences. They were not prompted by the interviewers to criticize the public system; however, the vast majority did, albeit in different manners and subtexts. Our analysis identified repetitive narrative patterns which indicate certain ‘key plots,’ meaning group narratives which include interpretation, value judgment, perceptual categories, and habitual patterns of action.

Following a long tradition of cultural-class studies starting in the 1960s and 1970s with Stuart Hall [51] in the United Kingdom, and with Pierre Bourdieu [52] in France, and inspired by Beverly Skeggs, we argue that these narrative patterns imply a cluster of strategies of interpreting and acting in reality. These strategies are collectively processed and narrativized in everyday interaction, finally becoming a ‘self’ in Skeggs’ terms (see above)—not a psychological self but a socio-cultural entity. Each individual might develop his or her unique variation of this collective cluster of strategies which is both the product of and the reaction to class conditions; both the result of determination and the locus of resistance [47,53].

The first type can be termed the ‘neoliberal patient-self’,’ which corresponds to the ‘hero quest’ narrative appearing in the stories of the upper middle-class. This patient-self presents itself as a reflective person making choices and using his or her high CHC to successfully navigate and easily cross the border between the public and the private systems, often using private insurance. The hero quest narrative is combined with unmistaken negative descriptions of the public system as inefficient and even hostile to the individual’s needs, arousing rage and contempt. This narrative involves consumerist discourse and meritocratic views. This seems to be a processed variation of an image produced by commercial insurance companies, HMOs, and leading policy makers who advance neoliberal views.

It is not surprising to see that this ideal self-manifestation adheres to the image of the white European middle-class male that Skeggs thoroughly examined [47], thus this type of neo-liberal patient self can appear in all societies in which public health care services have been partially commodified [54]. In Israel, it repeatedly appears in the form of the Jewish Ashkenazi upper-middle-class professional. The further we move away from this social type, if only a small step to lower status Ashkenazi women, the more the setbacks and potential contradictions in this ideal presentation become evident. One potential crack arises already in Ronny’s story from the fact that this single individual hero can seldom bridge the knowledge gap or the need for professional guidance resulting in increased risks being taken. From a professional perspective, Ronny’s back operation seems quite a hasty decision supported only by one doctor.

Turning to the narratives of the other ethno-classes, the public system continues to be presented negatively. Complaints range from inept and low professionalism, long waiting lines, lack of services, crowded hospital wards and indifferent or even humiliating bureaucracy. The majority of these descriptions were followed with a positive impression of private services, showing that the commodification of health has penetrated the middle-low and working classes. A closer look reveals important variations and distinctions between the other two patient-selves which attest to different ethno-class conditions.

A ‘troubled patient-self’ appears in the narratives of lower ethno-classes, which includes Mizrahi, Arab, and Russian respondents. The negative experiences of the public system are shared with the neoliberal patient self, but the tone is different. Instead of rage and contempt, the troubled patient self-expresses despair, fear, and feelings of abandonment. This patient-self usually embarks reluctantly or is pushed to search for alternatives, mainly using supplementary insurance.

Since supplementary insurance is significantly less expensive than commercial insurance, owning ‘just’ supplementary insurance attests to one’s position in the lower middle-class and working classes. Unlike the narrative of the neoliberal patient-self, the narrative of the troubled patient-self to ‘search for an alternative’ has a very mild feeling of success. Narratives strongly imply that this patient-self acknowledges that ‘things could have been different.’ Here the logic of care is still a vivid memory, longing for an alternative.

The tales of frustration, anger, and despair that emerged in the interviews stand in contrast, to a certain extent, to the findings of quantitative studies on the degree of satisfaction with the public healthcare system. Recent studies showed somewhat surprising levels of trust and satisfaction even among lower SES respondents [6,27]. For example, in one study 88% of the sample were satisfied with their HMO, and 68% reported being satisfied or very satisfied with the public healthcare system [3].

However, this seeming contradiction with our findings might be less dramatic. The two impressions might coexist. One might safely state that the myriads of experiences within the public system are not all as bad as the stories suggest. People are very complex creatures. This is a cliché, but it is our contention that respondents might answer a poll stating that the HMO is “all right”, while articulating a narrative that directs the processing of their experience in a different and negative manner.

The (cultural) fact in our study is that the recurrent narratives discuss hardships and barriers in the public healthcare system. It seems that a dominant image of a neoliberal patient-self-delivered by marketing, policy agents and paradoxically by the HMOs themselves—created a lasting and deep impact on both high and low ethno-class strata.

Construction of patient selves through narrative involves a selection of negative experiences rather than positive or neutral experiences to become the ‘key-plot’. Even the troubled patient-self, while it does not adopt the neoliberal hegemonic self in its full form, does accept its basic negative reading of reality in the public system.

The accounts of the troubled patient-self reflect the use of supplementary insurance—the insurance sold by the public HMOs. As mentioned above, these respondents probably tend to accept the blurred difference between the HMO’s public insurance and their private supplementary insurance. Hence, when using the latter, while expecting to get the benefits of the private service, they look at the (public) HMO as the primary source of care. A famous marketing slogan for one of the HMOs supplementary insurances claims that it provides ‘private quality with public prices’, blurring the distinctions while implicitly stating that the private sector is better than the failing public one. Hence, frustration with the public system has become a discursive common axiom, while the HMO plays a confusing game of private-public entity delivering mixed messages.

In summary, the troubled patient-self has internalized commodification or at least accepts its discourse as a natural fact. The narratives prioritize negative accounts of the experiences in the public sector and seek individual solutions using supplementary insurance. Yet, there are indications that the troubled patient-self is frustrated, angry, and fearful, more than contemptuous and strong, mostly seeking to survive, and it seems to hold some sense of the logic of care.

The third patient-self, referred to as the ‘communal alternative self,’ appears here uniquely in the narrative of the Arab respondents, mainly from the lower middle or working classes. Similar to the other two patient-selves, the narrative starts with negative descriptions of their experience in the public system. As with the troubled patient-self, the stories rarely express contempt for the public system. Rather, they express sorrow, anger and even humiliation, which is mostly associated with the behavior of providers in the field—doctors, nurses, and administrative personnel. Being a discriminated minority permeates these experiences.

However, instead of juggling between the public and the private systems using the power of money (private insurance), they construct a bypass route using the power of community networks of the Arab minority. The basic extended Arab family is widened so as to contact other networks through a chain of acquaintances among health providers who are all—or so they appear in the stories—Arab. At the center of the network lies its anchor, who is probably a local doctor, manager, or administrative, upon whom the patient can rely due to the intimate relationship of being a relative or at least a resident of the same village or town.

The patients might have reached their contacts through the public system, but the route leads them away informally to other public institutions or even to the private system. The formal barriers between institutions or between the public and the private systems seem to blur or vanish. The narratives start with long wait times, indifference, and even humiliation and discrimination in the public system. Then a chain of contacts is built on the spot, according to the specific needs. Monetary transactions might be involved as some doctors do expect a fee, yet this seems to be an ancillary issue, perhaps even slightly embarrassing, not an expression of commodification. It is (re)presented as a gift of gratitude. This way of navigating the system highlights pre-capitalist communitarian practices, compared to the other two patient-selves which show the constraints and influences of the neoliberal commodification of everyday life.

## 5. Conclusions

This research analyzed narratives told by patients from a variety of ethno-class backgrounds. The aim was to identify the strategies of interpretation and action that patients use when confronted with health needs in a context where neoliberal policies and culture have created a complex ‘maze’ of mixed public and private healthcare options.

The results portray a complex interplay between class determining power, where strategies are conditioned by ethno-class belonging, coupled with markers of group agency. The stories reflect a range of strategies of interpretation and action which consolidate into three social types, which we termed patient-selves. There is the neoliberal patient-self which is typically associated with the upper middle-class, especially Ashkenazi and male; the troubled patient-self, often representing the lower-middle and some working-class respondents, mostly Mizrahi and Arabs; and the ‘communal alternative patient-self’ which characterizes Arab respondents, mostly from a working-class background.

The neoliberal patient-self expresses negative impressions of the public system, with contempt and ridicule. This self uses its resources including private insurance and high CHC to maneuver successfully, building what we termed a hero-quest narrative. However, the success stories can hardly mask anxieties and real risks that are involved in the ‘quest’, which is carried out with poor knowledge and little professional coordination. These narratives discuss individual self-reliance, reflexiveness and use of CHC. This self has likely internalized the discourse issued by insurance companies and HMOs marketing strategies and by neoliberal policies. More generally, it is a reflection of the white Euro-American middle-class male which Skeggs examined thoroughly, as mentioned above.

The troubled patient-self begins with a negative perception of the public healthcare system, but the negative experiences told in these stories express sadness and fear rather than contempt or ridicule. Even if the narrative ends with a positive outcome for the patient, the overall tone is more of suffering than a celebration of individual choice. This patient-self refuses to, or maybe cannot, allow itself to abandon the expectation that the public system’s logic of care will reappear. It is commodified, but only up to a certain point. The feelings of this patient-self may represent some form of resistance, albeit in an individual and isolated manner.

Working- and lower middle-class Arabs share many experiences with the troubled patient-self, but some of them represent a different pattern—the ‘communal alternative’ patient-self. The networks of acquaintances enable the patient to bypass barriers. This communal care is based on the shared identity of a largely marginalized Arab minority. The fact that the Arab minority has integrated itself into health professions through individual mobility tracks [20,26] allows for this communal network to exist. This patient-self has adapted to neoliberal conditions and its discourses—the negative impressions of the public system and seeking individual solutions using personal resources to bypass its barriers—but the route itself is more subversive. Its commodified nature should be the subject of further inquiry.

The resulting picture is complex. The dominance of the neoliberal discourse has been internalized by the patients in Israel, as observed in other countries [55]. However, the two latter types also express a certain ‘voice’ or agency that resists neo-liberalization of the self as an all-encompassing progressive dynamic. The lower classes live in conditions of relative objective scarcity and subjective-cultural pressure (to conform with the neoliberal patient-self), hence any deviance from this route, even a small one, can be considered as an expression of agency and even resistance.

A final note might be required regarding the size of the sample and the validity of the claims. According to the literature, both on life-narrative and on grounded theory, there is no consensus on the size of the sample or on the exact point of “saturation”. As Bertaux and Kohli state:

“Some research projects are based on several hundred, others rely on a single one, and the majority fall somewhere in between. The number depends on whether empirically grounded generalization is being sought or whether one is using a case study approach, where only generalizations based on theoretical plausibility, not statistical induction, are possible.” [17].

In our study we do not claim that the coping strategies we found in the subjects’ narratives represent an exhaustive description of all possible stories. We It is far from our intention to argue that the coping strategies we found among our Israeli Arab interviewees or Mizrahim are the only coping strategies or that they determine each and every member of these ethno-classed groups. Rather, our aim is to offer exploratory insights or tentative social types that may serve as a basis for future research. This primary objective aligns generally with what Hammersley and Atkinson describe as an elaboration of theory using ethnography [56].

For example, among our Israeli Arab interviewees we identified a coping pattern that is different from the ones we found among Israeli Jews, and it has been hidden from other quantified studies. This is an example of a pattern which in our view is both subjective (as a classed self) and an objective result of the complex interaction between the exclusionary character of the Israeli society (regarding Arabs in this case) and the fact that the Israeli health care services is a universal one.

Lastly, it is important to stress that these patient-selves represent archetypes and the boundaries between them are fluid in real life. Class structure is not wholly deterministic, and individuals have a level of freedom to choose how to act with different options. For example, an Arab respondent struggling with upward mobility might adopt a neoliberal patient-self in his or her journey to participate in the upper middle-class. The patient-self is the complex interaction between the constraints of ethno-class structure and experiences, and individual and group agency.

### Future Research and Policy Change

Our research highlights the importance of incorporating ethno-class dynamics and patient agency into both academic inquiry and healthcare intervention design. Recognizing the distinct needs and coping strategies of various social groups can contribute to building more equitable and effective healthcare systems.

Future research should move beyond conventional satisfaction or trust surveys and adopt more nuanced, qualitative approaches that reflect the lived experiences of diverse ethno-class groups. Ethnographic and narrative methods, in particular, can reveal how patients construct their identities and develop strategies in response to structural barriers and neoliberal pressures. A deeper, intersectional analysis—one that accounts for the interplay of ethnicity, class, gender, and geography—is essential for capturing the complexity of healthcare experiences. Longitudinal studies could further illuminate how patient subjectivities evolve, particularly in response to changing policies or economic shifts. Comparative studies with countries experiencing similar processes of commodification would also help distinguish universal patterns from unique local dynamics [57].

Importantly, future research should explore how marginalized groups develop and express agency—not merely as individuals, but through communal networks and alternative pathways to care that often operate outside formal institutions.

From a practical standpoint, our findings suggest the need for both structural reforms and culturally sensitive communication strategies. Efforts should focus on helping all patients—especially those with lower levels of cultural health capital—navigate the healthcare system more effectively. The use of community-based health networks could be developed in non-commodifying ways, in order to combine decentralization with equity. Strengthening community-based health networks, particularly in Arab and other marginalized communities, could improve access and continuity of care by formalizing and supporting existing informal structures. Initiatives such as training local health mediators or system navigators could further support patients in overcoming institutional and bureaucratic barriers.

Addressing administrative, linguistic, and cultural obstacles within the public system is crucial, as is improving coordination between public and private providers to reduce fragmentation and confusion. Policy reforms should prioritize reducing out-of-pocket costs and expanding public funding to ensure that new services and technologies are distributed equitably across regions and social groups.

Promoting equity and social justice in healthcare requires a dual focus on material and symbolic inequalities. This includes fostering solidarity and a sense of collective responsibility that resists the individualizing logic of neoliberal discourse. Ultimately, both research and policy interventions must recognize and support the agency of patients from all backgrounds—empowering them to navigate, and at times resist, the increasingly commodified healthcare landscape.

## Figures and Tables

**Table 1 ijerph-22-01489-t001:** Ownership of Health Insurance in Israel (Adults 22+).

Insurance Type	1999	2001	2003	2005	2009	2012	2016
Supplementary insurance (Kupot Holim)	51%	64%	73%	79%	81%	81%	84%
Commercial insurance	24%	26%	35%	34%	35%	43%	57%
Both types (dual coverage)	13%	20%	28%	31%	32%	39%	52%

Source: Brammli-Greenberg et al. (2019) [31].

## Data Availability

The original contributions presented in this study are included in the article. Further inquiries can be directed to the corresponding author.

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
