# Peer review of "Commodification of Healthcare—Patient Perspective: A Cultural-Class Inquiry of Patients’ Experience in Public–Private Systems in Israel"

_ijerph, 2025, doi:10.3390/ijerph22101489_

Round 1
Reviewer 1 Report (Previous Reviewer 1)
Comments and Suggestions for Authors
I really appreciate how you have responded to questions with the draft and I think the paper is much stronger and easier to follow. A couple of additional suggestions:
1) I do not think that you have described the process of obtaining study participants in sufficient detail. I do not think that you have provided enough detail about your sampling to allow for replication of the study.
2) I think you need to reference the timeframe of this study.....Was data collected before the October 7 attack and Israeli military response or since then? I think this matters and should be stated.
3) I appreciated your reference to a second group of respondents to learn their thoughts on the findings. Who were these people and how were they selected and interviewed? Did you explicitly ask them for counter-examples and alternative views?
4) You do not provide a strong rationale for believing in the reliability of data gathered from you sample. You do need to acknowledge that the sample members' experiences may or may not be representative unless for example you explored thematic saturation or some other method to make sure that your sample and resulting themese are reliable takes on the whole population.
5) I thought there was a bit of repetition of key methods/theory concepts in this version, I would take one more sweep through to look for opportunities to streamline the text.
6) In reading the experiences attributed to working class Arabs in your results section, I had trouble sorting out how these experiences were caused by blatant cross-group prejudice (racism) and social class prejudice on the part of health professionals vs. limited capacity in the public system....some comments clarifying this would be helpful.
Author Response
Please check the attached file.

Reviewer 2 Report (Previous Reviewer 3)
Comments and Suggestions for Authors
The authors have done a tremendous job of revising this paper. I enjoyed reading it.
(I encourage them to move ahead with their suggested revision of conceptual frameworks based on choice!)
Author Response
Thank you for your comments.
This manuscript is a resubmission of an earlier submission. The following is a list of the peer review reports and author responses from that submission.
Round 1
Reviewer 1 Report
Comments and Suggestions for Authors
This paper explores some important issues around how the commodification of health care and under-investment in health care in Israel result in diverse patient experiences based on the intersections of economic status and ethnic group in the context of Israel's apartheid state.
This small sample qualitative study does not report on the use of international standards for the conduct and reporting of qualitative research (see for just one example https://www.equator-network.org/reporting-guidelines-study-design/qualitative-research/ or https://pubmed.ncbi.nlm.nih.gov/24979285/) and would not thus not meet the quality standards expected by readers of this journal.
This study includes 4 Arab Israelis and draws conclusions on health related informal social structure in this complex community.....this is just offensive....There is no discussion of how limiting the small sample of Arab Israelis might be....there is no discussion of thematic completeness and sample representativeness.
What I found particularly troubling about this paper, despite the selection of traditionally left wing analytic categories ("ethno-class, focusing on how shared situations shape individual experiences for a group of people as an indicator of systemic oppression) and yet only very briefly mentioning the context of the ongoing war on the Palestinian people and the associated disruption throughout much of the country even prior to October 7. It seems unlikely that all of the complex feelings of entitlement and endurance and despair that you describe could be unrelated to this borader context. Similarly, you seem to exclude any discussion of how racial-ethnic differences play out in delivery and reciept of private and public care: and how that influences experiences of being cared about and treated well. Overall, by focusing on deriving the three archetypes, the analysis becomes decontextualized in exactly the ways that your critiques of prior research noted as a concern.
Finally, although it is not a requirement, readers of this journal would like to hear your thoughs about how the findings of your research can be applies in improved research and practical interventions.
Reviewer 2 Report
Comments and Suggestions for Authors
This manuscript is a compelling and original contribution to the sociological and public health literature on the commodification of healthcare. Through a well-executed narrative analysis of 20 in-depth interviews, the authors critically examine how different ethno-class patient groups in Israel navigate the increasingly commodified healthcare landscape. The theoretical framing—drawing on Skeggs, Bourdieu, and Mol—is sophisticated, and the concept of “patient-selves” (neoliberal, troubled, and communal-alternative) offers a rich typology that enhances understanding of both individual agency and structural constraint. The manuscript is well-written, methodologically robust, and situated in relevant international literature.
However, some revisions could further strengthen the clarity, rigor, and impact of the article.
- Significance and Originality: The article fills a critical gap in the literature on health system commodification by grounding patient narratives within ethno-class structures, particularly in the Israeli context. The integration of cultural sociology and public health perspectives is innovative and generative. The typology of patient-selves provides both theoretical and practical implications for healthcare equity, particularly in stratified and mixed public-private systems.
Suggestions:
- Consider more directly linking the Israeli case to global debates on commodification (e.g., by referencing OECD comparisons, or health system typologies).
- While the theoretical engagement is rich, the contribution to policy debates could be more explicitly stated in the conclusion.
- Theoretical Framework: Excellent use of Beverly Skeggs' concept of the “classed self,” with a critical engagement that avoids simplistic application of Bourdieu. Strong integration of other relevant theorists (Mol, Shim, Harley, Hall, etc.).
Suggestions:
- The critique of Mol’s overemphasis on “choice” is valid, but the manuscript might benefit from specifying how “subjectification” expands the analysis (e.g., through affect, temporalities, or informal systems of care).
- Further clarification on how the “communal alternative self” interacts with or resists neoliberal subject formation would enhance the theoretical depth.
- Methodology: Narrative analysis is an appropriate and effective choice given the study's focus on meaning-making and lived experience. The ethical considerations and sampling strategy (snowballing, regional diversity, gender, ethnicity) are well documented.
Suggestions:
- Provide more information on saturation and whether 20 interviews were sufficient to identify key themes.
- Expand briefly on how themes were coded in ATLAS.ti—what was the coding process, and how was intercoder reliability ensured (if applicable)?
- Consider including more demographic data or a brief table summary earlier in the paper (not just in Appendix A).
- Results and Interpretation: The patient narratives are powerful and richly presented. The identification of the three “patient-selves” is insightful and sociologically rigorous.
Suggestions:
- Consider integrating brief subheadings for each narrative example for easier navigation (e.g., "Ronny: The Hero Quest").
- The voices of Arab participants—especially around community-based navigation—are particularly novel and important. Consider elevating their insights into a broader discussion of cultural resilience or solidarity economies.
- Discussion and Conclusion: The discussion provides a thoughtful interpretation that resists simplistic binaries (e.g., public/private, empowered/disempowered). The concept of resistance through informal networks is especially compelling in relation to structural inequality.
Suggestions:
- Add a clearer articulation of the implications for health policy and equity: What could health systems learn from the communal alternative patient-self?
- Strengthen the international relevance of the typology by reflecting on whether these patient-selves might appear in other commodified systems (e.g., the U.S., Australia, UK).
- References and Citations: Comprehensive, well-integrated use of interdisciplinary scholarship. Good balance between Israeli-specific and international literature.
Suggestions:
- Ensure that key recent literature on healthcare equity, especially in the Global South or among racialized populations in the U.S. and UK, is not overlooked (e.g., Farmer, Navarro, or more recent critiques of neoliberalism in health systems).
- Minor Comments and Edits
- Ensure consistent use of acronyms and definitions (e.g., HMO, CHC, LMC).
- Line edits: "Selfs" should be corrected to “selves” throughout.
- Some quotes could be tightened for clarity and conciseness while preserving voice.
Reviewer 3 Report
Comments and Suggestions for Authors
This paper addresses an interesting question -- how class and ethnicity intersect to shape help-seeking behavior and attitudes in an increasingly privatized healthcare system. Some of the findings are interesting, e.g., the informal networking of Arab Israelis. However, in its current form, there are three primary problems: the need for a more straightforward explanation of the theoretical framework, an absence of some important details about the Israeli healthcare system, and a concern experienced by this reader that the results were shaped to fit the starting hypotheses.
Theoretical framework
I recommend that after the introductory paragraph, the authors focus exclusively on their theoretical framework, then move to explain the context of their study, i.e., the Israeli context. In discussing the theoretical framework, the goal should be to clearly state their approach to this work (as opposed to giving a full treatise on the relevan theoretical literature) and to do so in a way that is accessible to an audience of non-sociologists. This is currently not the cause. For example:
-When the authors first mention Bourdieu, it is to tell us that "Jannette Shim's analysis is perhaps the most extensive theoretical elaboration of Bourdieu's terminology." But they haven't said anything about Bourdieu to this point. What they have been saying may be Bourdieu's "terminology," but we don't know that. Similarly, when they later say that they are aware of Bourdieu's field theory and explain why they are not using it, they seem to assume that the reader understands what that is. That assumption would not be advisable even for a sociology journal, but this journal is not that.
-After explaining their concerns about Bourdieu (although the reader hasn't been given the full information to understand that), they then talk about Skeggs, leaving one to wonder why they have not quickly explained what Bourdieu does and does not contribute to their framework and moved on to Skeggs.
They make this comment about their framework:
The present study is informed and inspired by the above scholars and research, but some fine theroretical adjustments must be made to account for the Israeli situation as we observe it. Specifically, we add two requirements that we believe contribute to understanding the specific ways in which class and ethnicity interact with the commodification of health care. Firstly, our question is more general than the issue of choice alone, addressing the ways in which experiences construct subjectivities as users/consumers of health care. Secondly, we assume that complex interactions in each social group construct not only the individual patient's CHC but the whole ethno-class repertoire of meanings and actions in complex health systems.
This raised a number of questions for me: 1) They say that their adjustments were needed to make this apply to the Israeli context. But their two adjustments don't seem particularly unique to the Israeli context to me. Also, if the theory doesn't account for a particular context, then it needs adjustment more generally. 2) Were these a priori adjustments they made before conducting their research or did these insights arise from the research? 3) Is this theoretical adjustment the critical point of this article? At times feel like it is and other times feels just like an aside. If it is, needs to be more consistently addressed as such.
Israeli Context
There is some very nice explanation of what ethno-classes look like in the Israeli context, including how they have changed, but other information is more scant or missing. Specific examples:
1) The authors assert a history of neoliberal reforms of the healthcare system in recent decades, but no examples are provided.
2) Figure 1 doesn't make the case that the authors are trying to make in the narrative. Lines are basically flat with a spike only in the public sphere at one point, which seems to undercut their argument.
3) I was confused by the differentiation between supplementary HMO insurance and commercial insurance, as in the United States HMOs are commercial. At the very end of the article, there is mention of HMOs being public. This requires explanation up front.
4) Data on changes in the percentage of people with public, complementary and commercial insurance would be helpful.
Results
It's fine to say, as the authors do here, that "an individual account of a seemingly isolated personal event might imply a grain of a 'key plot,' or a collective 'story' that encapsulates the experience and repertoire of the whole group." However, there was something about the lead up to the results section that gave me the feeling that the authors were intent on believing that they would get consistent stories within each group and the presentation of the results themselves seemed to indicate a significant bias towards seeing what they wanted to see. For example:
-One respondent who indicates their privileged position of having non-public financing ("I’m telling you – I went to a gastro [private specialist]. I hardly sat down, and he already called my name") is commented on this way: "Yet her tale is not about private medicine’s victory." It seems to me that it does.
-Yossi and Ja'far are described as neither mocking nor showing contempt for the public system and yet they used words like "disgusting" to describe their experiences there.
-Arabs who use personal networks to get care are described as using "coping strategies," whereas when Ronny goes with a private doctor whom he meets, he is described as seeing himself to be on a hero's quest.
The distinctions the authors are making may be valid but the way the findings are currently presented makes the reader question their interpretation of their results. It is not clear from the methods that standard approaches to increasing the validity of qualitative research have been employed, e.g., blinded double coding, member checking, self-reflection, etc. (Also, the questions have not been provided to give us context for the responses.) The authors should employ at least one of the standard validation methods and also be as clear as possible about any distinctions they continue to find valid.
-
